# Impact of Mediterranean Diet Adherence During Pregnancy on Preeclampsia, Gestational Diabetes Mellitus, and Excessive Gestational Weight Gain: A Systematic Review of Observational Studies and Randomized Controlled Trials

**DOI:** 10.3390/nu17101723

**Published:** 2025-05-20

**Authors:** Sukshma Sharma, Simona Esposito, Augusto Di Castelnuovo, Alessandro Gialluisi, Paola De Domenico, Giovanni de Gaetano, Marialaura Bonaccio, Licia Iacoviello

**Affiliations:** 1Research Unit of Epidemiology and Prevention, IRCCS Neuromed, 86077 Pozzilli, IS, Italy; research.sukshma@gmail.com (S.S.); simona.esposito@moli-sani.org (S.E.); dicastel@moli-sani.org (A.D.C.); alessandro.gialluisi@gmail.com (A.G.); giovanni.degaetano@moli-sani.org (G.d.G.); licia.iacoviello@moli-sani.org (L.I.); 2Department of Medicine and Surgery, LUM University, 70010 Casamassima, BA, Italy; 3Istituto Clinico Mediterraneo, 84043 Agropoli, SA, Italy; paoladedome@gmail.com

**Keywords:** Mediterranean diet, pregnancy, maternal diet, preeclampsia, diabetes mellitus, weight gain, risk factors: protective factors

## Abstract

Background/Objectives: There is limited evidence on the association between maternal Mediterranean diet (MD) adherence and risks of preeclampsia, gestational diabetes mellitus (GDM), and excessive gestational weight gain (eGWG), and hence a systematic review of observational studies and randomized controlled trials was conducted. Methods: A total of 30,930 articles from the Scopus, EMBASE, PubMed, MEDLINE, and Google Scholar databases were identified, published between January 2000 and April 2025. The National Institutes of Health Quality Assessment Tool and the Cochrane Risk of Bias Tool 2.0 were used to assess the quality of seven studies (one each were case-control and cross-sectional, three were RCTs, and two were prospective cohort studies). Results: Overall, the studies examined the risks of preeclampsia (four studies), GDM (five studies), and eGWG (three studies). Only one prospective cohort study out of four reported that MD adherence was associated with lower risk of preeclampsia (OR 0.78; 95% CI: 0.64 to 0.96 for highest vs. lowest tertile). MD adherence was associated with decreased risk of GDM in the intervention groups in four studies (two RCTs (OR: 0.75, 95% CI 0.57 to 0.98 and OR: 0.72, 95% CI 0.50 to 0.97) and one each of cross-sectional (OR: 2.32; 95% CI 2.13 to 2.57 for a 1-point decrease in the dietary score) and case-control studies (high Vs. low MD adherence: OR: 0.20, 95% CI 0.50 to 0.70)). MD adherence was associated with decreased risk of eGWG in two studies: one RCT (RR: 0.91, 95% CI 0.86 to 0.96 for a 1-point increment in the MD score) and one cross-sectional study (OR: 1.78; 95% CI 1.51 to 2.02 for a 1-point decrease in the MD score). Conclusions: Findings indicated the protective associations between MD adherence and GDM and eGWG risks but not for preeclampsia.

## 1. Introduction

During pregnancy, a woman’s body undergoes a series of dynamic and significant physiological changes to adapt and accommodate the optimal development of the growing fetus, including increased blood volume, accumulation of fat reserves, and alterations in glucose, protein, and calcium metabolism [1,2]. Optimal maternal nutrition during pregnancy is a major contributor to healthy pregnancy outcomes [3,4,5,6,7,8]. Previous nutritional evidence has suggested the protective effects of single-nutrient intakes (for instance, vitamin B12, folate, iodine, and iron), maternal diets and dietary patterns (including “plant-based dietary patterns” and “prudent or traditional diets”) on healthy pregnancy outcomes (longer pregnancy duration, reduced risks of GDM and preterm birth, healthy gestational weight gain, lower risks of anemia, and reduced maternal mortality) [3,9,10,11,12], and healthy offspring growth (adequate birth weight and low risks of small-for-gestational-age (SGA) and preterm birth) [3,13,14,15,16,17,18,19,20,21,22,23,24,25,26,27,28].

The protective effects of MD on human health, including cardiovascular health, cancer, and obesity, were not globally recognized until the 1990s [29,30]. The traditional MD is characterized by high intakes of vegetables, fruits, cereals, nuts, and legumes, moderate intakes of dairy products, fish, and wine during main meals, and low intakes of red meat, meat derivatives, and use of olive oil as the main source of added fat [31,32]. It has been extensively explored for its high polyphenol content, exhibiting anti-inflammatory and antioxidant properties [33,34,35,36]. To date, systematic reviews have suggested an inverse association between MD adherence and maternal outcomes, including preterm deliveries, pregnancy-induced hypertension, urinary tract infections, and maternal gut microflora, as well as poor birth outcomes, including low birth weight, intrauterine growth restriction, and SGA and large-for-gestational-age (LGA) deliveries [37,38,39,40].

Higher prevalence rates of preeclampsia, GDM, and eGWG in pregnancy [41,42,43] reflect the increasing prevalence of obesity and changes in women’s lifestyle. The National Institute of Child Health and Human Development and the Centers for Disease Control and Prevention have reported GDM, eGWG, and preeclampsia as the most common pregnancy complications [44,45]. The World Health Organization (WHO) reported global prevalence rates for preeclampsia (2% to 10%) and obesity during pregnancy (eGWG) (1.8% to 25.3%). Furthermore, based on the International Association of Diabetes and Pregnancy Study Groups (IADPSG) criteria, the global prevalence of GDM is 14.7% [41,46,47]. Studies have suggested that preeclampsia and GDM share similar risk factors, including eGWG, advanced age, and multiple pregnancies [48,49,50], with similar pathophysiological processes involving oxidative stress, proinflammatory release factor, and vascular endothelial dysfunction, therefore increasing risks of future cardiovascular disease and diabetes [50].

However, previous reviews have reported limited (regarding the maternal outcomes) and inconclusive findings on the association between MD adherence and the risks of preeclampsia, GDM, and eGWG, due to methodological inconsistencies and study limitations [37,38]. This has highlighted the requirement for clear and comprehensive systematic reviews that provide up-to-date evidence regarding the general population of well-nourished pregnant women useful in clinical and public health settings [51,52,53,54]. We conducted a systematic review of observational studies and randomized controlled trials (RCTs) to examine the association between MD adherence during pregnancy and preeclampsia, GDM, and/or eGWG.

## 2. Materials and Methods

### 2.1. Eligibility Criteria, Information Sources and Search Strategy

Table 1 shows the eligibility criteria for the inclusion of studies according to the PICOS framework (Participant, Intervention, Comparators, Outcomes, and Study design). This systematic review included studies that examined the associations between MD adherence during pregnancy (exposure) and risk of preeclampsia, GDM, and eGWG (outcomes), with the following study designs: (1) observational studies (case-control, cross-sectional and cohort studies), and (2) RCTs to limit potential biases and maximize confidence in estimates (including parallel, cross-over, and factorial designs). Only human studies published in English with adult participants (>18 years of age) were included in the systematic review. Also, only studies that conducted research in a general population of pregnant women were included, and not studies that examined women with metabolic risk factors and/or diseases (for example, obesity, hyperlipidemia, hypercholesterolemia, chronic hypertension, or Type I or II Diabetes Mellitus).

The following types of study were excluded: abstracts, case reviews, case reports, editorials, magazine articles, preprint publications, newsletters, systematic reviews, umbrella reviews, letters to the editor, thesis material, dissertation chapters, surveys, literature reviews, studies conducted on animal models, laboratory (in vitro) studies, non-RCTs, and conference abstracts. Further, studies that explored other types of diets, dietary patterns, and diet indexes as study exposures or focused on healthy diets or different types of diets consumed before pregnancy/pre-conception diets were excluded.

Also, outcomes, including minor aged (teenage) pregnancies, offspring outcomes, fetal complications, and maternal complications other than the above-mentioned conditions i.e., preeclampsia, GDM and eGWG were excluded. Finally, studies published in non-English languages were excluded.

A systematic search was initially conducted in September 2022 and was then repeated in April 2025 of 5 databases (Google Scholar, EMBASE, Medline, Scopus, and PubMed), with publication date restrictions set between 2000 and 2025 to examine the recent literature related to MD and maternal outcomes. Our literature search on multiple databases, including Google Scholar, Scopus, and EMBASE (PubMed and Medline), indicated that studies on maternal outcomes, including preeclampsia, GDM, and eGWG, only emerged around the early 2000s. Hence, we decided to evaluate the evidence from 2000 to 2025.

Searches were performed using predefined key words related to pregnancy complications, Mediterranean diet adherence, pregnancy outcomes, delivery outcomes, maternal complications, preeclampsia, gestational weight gain and gestational diabetes mellitus, with MeSH terms in different combinations wherever applicable, such as the following: ((‘mediterranean’/exp OR mediterranean) AND (‘diet’/exp OR diet) AND (‘pregnancy’/exp OR pregnancy) AND (‘outcomes’/exp OR outcomes) OR ‘maternal’/exp OR maternal) AND (‘complications’/exp OR complications) AND (‘delivery’/exp OR delivery) AND (‘outcomes’/exp OR outcomes) AND ([embase]/lim OR [medline]/lim OR [pubmed-not-medline]/lim), (‘mediterranean’/exp OR mediterranean) AND (‘diet’/exp OR diet) AND in AND (‘pregnancy’/exp OR pregnancy) AND (‘preeclampsia,’/exp OR preeclampsia,) AND (‘diabetes’/exp OR diabetes) AND mellitus AND gestational AND (‘weight’/exp OR weight) AND (‘gain’/exp OR gain) AND ([embase]/lim OR [medline]/lim OR [pubmed-not-medline]/lim) AND [2022–2025]/py, (‘mediterranean’/exp OR ‘mediterranean’ OR ‘mediterranean’/exp OR mediterranean) AND (‘diet’/exp OR ‘diet’ OR ‘diet’/exp OR diet) AND in AND (‘pregnancy’/exp OR ‘pregnancy’ OR ‘pregnancy’/exp OR pregnancy) AND (‘preeclampsia,’/exp OR ‘preeclampsia,’ OR ‘preeclampsia,’/exp OR preeclampsia,) AND (‘gestational diabetes’/exp OR ‘gestational diabetes’) AND (‘diabetes’/exp OR ‘diabetes’ OR ‘diabetes’/exp OR diabetes) AND mellitus AND (‘pregnancy’/exp OR pregnancy) AND (‘weight’/exp OR ‘weight’ OR ‘weight’/exp OR weight) AND (‘gain’/exp OR ‘gain’ OR ‘gain’/exp OR gain) AND ([embase]/lim OR [medline]/lim OR [pubmed-not-medline]/lim) AND [2022–2025]/py (refer to Appendix A for a full search strategy performed within each database). The reference lists of all publications were also searched for potential studies that could be included in the current systematic review. Finally, the full-text versions of the extracted articles were stored as PDF files and managed using Mendeley Reference Manager version 2.79.0.

The current systematic review was performed according to the Preferred Reporting Items for Systematic reviews and Meta-Analyses (PRISMA) guidelines [55] (see PRISMA checklist in the Appendix A). This systematic review was registered at PROSPERO https://www.crd.york.ac.uk/prospero/display_record.php?ID=CRD42023440494 (accessed on 5 May 2025).

### 2.2. Study Selection

The details of the identified records retrieved from databases were then transferred onto an MS Excel spreadsheet for the screening process. Research articles were screened based on title and abstract by 2 reviewers (SE and SS). The ‘Find’ function in the Excel spreadsheet helped to identify duplicates and eliminate duplicate entries from all the databases. One reviewer (SS) conducted a final check to reduce selection errors. The full-text versions of the studies were retrieved and evaluated for inclusion based on the inclusion criteria elaborated above—two reviewers assessed them (SE and SS), and a third reviewer verified the same (MB). The screening process is provided using the PRISMA flowchart, version 2020 [55] (see Figure 1).

### 2.3. Data Extraction

The following data were extracted from the included studies: author, publication year, study design, study population, country, sample size, gestational weeks at recruitment, dietary assessment used for dietary data, measurement of Mediterranean diet adherence (study exposure), covariates, statistical methods, other outcomes of interest in the study published, assessment of study outcomes (preeclampsia, GDM and/or eGWG), and results (Table 2 and Table 3). The following data were added for randomized intervention trials: a sample size of the control and intervention group and nutritional supplementation provided to the interventional group (Table 2).

Two independent reviewers (SE and SS) assessed the extracted data and selected the studies. One author (SE) extracted data related to the sample, methods, and results from individual studies and cross-checked them by one reviewer (SS). Studies were included only upon the mutual agreement of both reviewers (SE and SS), and differences of opinion, if any, were solved by a third reviewer (MB), wherever necessary.

### 2.4. Definitions of Outcomes

The definitions for the three outcomes, namely, preeclampsia, GDM, and eGWG were as follows: (A) Preeclampsia was defined according to the American College of Obstetrics and Gynecology (ACOG) [63]: (1) the presence of hypertension (systolic > 140 mmHg or diastolic > 90 mmHg, on 2 occasions, 4 h apart) in previously normotensive women, and (2) proteinuria (>300 mg/24 h urine collection or protein/creatinine > 0–3 or dipstick reading = 1+), or (3) severe features, including thrombocytopenia (>100,000 mcgL), pulmonary edema, or new-onset cerebral or visual symptoms [48]. (B) Two definitions were used for GDM: (1) the International Association of Diabetic Pregnancy Study Group (IADPSG) recommends that fasting plasma glucose ≥ 5.1 mmol/L at early pregnancy be diagnosed as gestational diabetes mellitus, and (2) the American Diabetes Association (ADA) guidelines recommend the use of a one-step 75 g oral glucose tolerance test (OGTT) (mmol/L) with IADPSG criteria between 24 and 28 gestational weeks in low-risk pregnant women [64,65,66]. Finally, (C) GWG was calculated as the difference between the weight at the first trimester and the last prenatal visit before delivery [67,68]. For eGWG, body mass index (BMI)-specific guidelines as per the 2009 Institute of Medicine (IOM) guidelines were used to determine the adequacy of GWG, listed as four categories [67]: (1) underweight (TGWG 12.5–18 kg), (2) normal weight (TGWG 11.5–16 kg), (3) overweight (TGWG 7–11.5 kg), and (4) obese (TGWG 5–9 kg) [68].

### 2.5. Assessment of Risk of Bias

Two risk-of-bias tools were used to assess the quality of the included studies in this systematic review: the National Heart, Lung and Blood Institute (NIH) Quality Assessment Tools for observational studies and the Cochrane Risk-of-Bias tool for randomized controlled trials or cluster-controlled randomized trials (RoB 2.0 tool) [69].

The NIH Quality Assessment Tools [70] were used to assess the risk of bias for observational cohort and cross-sectional studies (14-item criteria) and case-control studies (12-item criteria). This tool did not use a points-based system; instead, it used three categories for methodological quality based on the overall judgment of the study: “Good”, “Fair”, and “Poor”. Two evaluators (SS and MB) assessed the quality of the included studies, and disagreements were resolved by discussing relevant parts of the paper.

Evaluators (SS and MB) had to select “Yes”, “No”, or “Not Reported/Not Applicable/Unable to Determine” in the NIH tool. An overall assessment for each study was generated based on the number of times “Yes” was selected under each criterion of the NIH tool; a “Good” study had a maximum of 3 categories that were not rated as a “Yes”. Two categories, “validity of outcomes” and “adjustment of confounders” were considered the most important criteria to determine the classification of study quality.

The RoB 2.0 tool [69] had 5 domains to assess the study: randomization process, deviations from intended interventions, missing outcome data, outcome measurement, and selection of the reported result. Each domain was used to assess the risk of bias. Once assessed, the studies were classed into three categories: (1) “low risk” of bias when a low risk of bias was determined in all five domains; (2) “some concerns” if >1 domain was assessed as raising some concerns but did not have a “high risk of bias” for any of the individual domains; and (3) “high risk” of bias when high risk of bias was assessed for >1 domain or the study assessment included “some concerns” amongst multiple domains [69]. The quality assessment results of studies examining MD adherence in pregnancy in relation to preeclampsia, GDM, or eGWG are summarized in Appendix A.

## 3. Results

### 3.1. Study Selection Findings

Overall, 30,930 records were identified from Google Scholar, Scopus, and EMBASE (MEDLINE and PubMed) databases that explored the association between maternal adherence to MD and pregnancy and birth outcomes. After removing duplicates, 30,928 studies were screened based on title and abstract. A total of 30,921 records were excluded for the reasons specified in the PRISMA flowchart (refer to Figure 1). Finally, full-text versions of the seven studies that explored the association between MD adherence in pregnancy and risks of preeclampsia, GDM, and/or eGWG were retrieved and included in the current systematic review.

### 3.2. Characteristics of the Included Studies

Of the seven studies, three were randomized controlled trials (RCTs) [60,61,62], two were prospective cohort studies [57,58], and one study each was of case-control [59] and cross-sectional study design [56]. All were published between January 2000 and January 2024 (for further details, refer to Table 2).

Amongst the seven studies, five studies examined the risk of GDM [56,57,59,60,62], four studies examined the risk of preeclampsia [57,58,61,62], and three studies examined the risk of eGWG [56,61,62]. The maternal age across all seven studies ranged between 23 and 35 years, and four studies categorized maternal ethnicity as Hispanic, African American, and Caucasian [58,60,61,62].

The RCTs included women in early pregnancy (8 to 12 weeks of gestation), whereas the prospective cohort, cross-sectional, and case-control study designs included pregnant women between 5 and 38 weeks of gestation. Of the three study outcomes, incidence of preeclampsia ranged from 1% to 10%, GDM from 5% to 43%, and eGWG from 18.2% to 43.4%.

Of the three RCTs included in the current systematic review, one RCT [60] was the original study, namely the St. Carlos Gestational Diabetes prevention study in Spain, which explored GDM risk, and the other two studies were conducted within the same trial as a part of the secondary analysis [61,62]. The sample sizes ranged between 600 and 932 participants divided into control and interventional groups according to the respective study design and objective.

All participants were recruited between 8 and 12 weeks of gestation, and the IADPSG criteria were used to measure gestational blood glucose levels and ascertain GDM. Although two studies used self-reported 7-day food diaries [60,61], and the third study used two self-reported, semi-quantitative FFQs [62], all three studies commonly used a 14-point MEDAS tool to assess participants’ adherence to MD.

The dietary advice for the RCTs’ interventional and control groups differed depending on the study objectives. The authors of the RCTs did not provide any diet plan or food to the participants in the interventional group. Furthermore, the participants were only advised to adhere to an MD and increase their intake of extra virgin olive oil (up to 40 mL/day) and nuts (pistachios), whereas the control group’s participants were advised to consume a standard diet (or one of limited fat intake). Amongst outcomes, two studies each explored the risks of eGWG and preeclampsia [61,62], and GDM [60,62].

Amongst observational studies, two prospective cohort studies were conducted in the USA [57,58], were established between 1998 and 2016, and ranged between 1887 and 8507 participants. Participants were recruited from the general population [57] and medical centers [58]. The Minhas study used a 16-item FFQ from a larger FFQ and Mediterranean-style diet score (MSDS) score (4–38), whereas the Li et al. study used a 124-item FFQ and the aMED score (0–9) to examine adherence to the MD. Amongst the study outcomes, both studies examined the risk of preeclampsia, but Li et al. also examined the risk of GDM. For preeclampsia and GDM outcome ascertainment, the Li et al. study used definitions by the ACOG 2002 criteria (for preeclampsia) and Carpenter–Coustan criteria (for GDM). However, the Minhas et al. study used a generalized definition to ascertain the risk of preeclampsia.

Amongst cross-sectional and case-control studies, the studies were conducted in Greece [56] and Iran [59], respectively, and comprised 463 to 5688 participants recruited from an online survey and hospital centers [56,59].

For dietary intake and measurement of MD adherence, the cross-sectional study [56] used a MediDiet Score questionnaire comprised of 11 food groups and used the MediDiet score (ranging from 0 to 55) by Panagiotakos et al. [71], whereas the case-control study [59] used three 24 h diet recalls to assess dietary intakes and MedDiet score (ranging 0–9) by Trichopoulou et al. [31]. Amongst study outcomes, both the cross-sectional study [56] and the case-control study used a general definition of GDM [59]. For eGWG, the cross-sectional study used the WHO definition of eGWG [56].

### 3.3. Results of the Included Studies

The results of the association between high MD adherence during pregnancy and the risk of preeclampsia, GDM, and eGWG in the included studies differed regardless of their study design (refer to Table 3 for detailed results). Finally, the findings were synthesized according to each study design, and results from only models fully adjusted for covariates were used (refer to Table 3 for the list of covariates used in each study).

Among the four studies that examined preeclampsia as a study outcome, only one study [58] reported an inverse association between high MD adherence during pregnancy and risk of preeclampsia (odds ratio [OR] 0.72; 95% confidence interval [CI]: 0.59 to 0.89 for T3 high vs. T2 medium MD score adherence) and (OR 0.78; 95% CI: 0.64 to 0.96 for T3 high vs. T1 low), whereas the other three studies reported no evidence of an association [57,61,62].

Four [56,59,60,62] out of the five studies that examined the risk of GDM reported an inverse association between high MD adherence in pregnancy and low risk of GDM, but not in the US cohort study [57].

To elaborate, an inverse risk of GDM associated with adherence to an MD was reported in one cross-sectional study (OR: 2.32; 95% CI 2.13 to 2.57 for a 1-point decrease in the dietary score) [56], one case-control study (OR: 0.20, 95% CI 0.50 to 0.70 of T3 high vs. T1 low of MedDiet score adherence) [59], and two RCTs (OR: 0.75, 95% CI 0.57 to 0.98 in the intervention group of a 1-point increase in the MEDAS score supplemented with extra virgin olive oil and pistachios), and (RR: 0.72, 95% CI 0.50 to 0.97) in the intervention and (RR: 0.77, 95% CI 0.61 to 0.97) in the real world group [RWG]) [60,62], respectively.

Finally, for eGWG, three studies reported that a high MD adherence was associated with a decreased risk of eGWG in an RCT (RR: 0.91, 95% CI 0.86 to 0.96 for a 1-point increment in the MD score) [61,62], while low MD adherence was associated with an increased eGWG risk in a cross-sectional study (OR: 1.78; 95% CI 1.51 to 2.02 for a 1-point decrease in the MD score) [56].

All studies analyzed the association between adherence to an MD and risk of preeclampsia, GDM, and eGWG using multivariable logistic regression models and most commonly adjusted for maternal age, parity, body mass index, and socioeconomic status except for one study (one RCT) [61].

### 3.4. Risk of Bias in Included Studies

The NIH tool for observational studies was used for three studies included in the systematic review (refer to Appendix A for a summarized version of the risk-of-bias assessment). Four studies (two studies of prospective and one each of cross-sectional and case-control study designs) scored “Good” overall, showing the included studies to have low risk of bias.

Furthermore, the RoB 2.0 Cochrane risk-of-bias assessment tool was used [69] to assess the risk of bias in three RCTs and scored overall “High Risk”. The three RCTs were poorly rated in the following domains: randomization process and selection of the reported result.

## 4. Discussion

The main objective of this systematic review was to investigate the evidence from observational studies and RCTs evaluating the associations between adherence to an MD in pregnancy and the risk of preeclampsia, GDM, and/or eGWG. Findings from all studies but one suggested that high adherence to MD in pregnancy was associated with a lower risk of GDM and eGWG. In contrast, all studies but one suggested no evidence of an association with the risk of preeclampsia.

Among the four studies that examined preeclampsia as a study outcome, only one study [58] reported an inverse association between high MD adherence during pregnancy and risk of preeclampsia. This might be due to any of three reasons: (1) different MD adherence scales used—RCTs used the MEDAS tool as compared to the MSDS tool used in the observational study; (2) preeclampsia cases were more frequent in the observational study (10%) as compared to the RCTs (1–3%), increasing the power to detect associations in the regression models [72,73]; and (3) the duration of the intervention in the RCTs could have been too short to observe any effects. Therefore, large sample-sized studies using a consistent definition of outcome could yield more evidence in relation to preeclampsia. Four [56,59,60,62] out the five studies that examined the risk of GDM reported an inverse association between high MD adherence in pregnancy and low risk of GDM. This might be due to the use of a different definition for GDM ascertainment (Carpenter–Coustan criteria) as compared to the commonly used IADPSG criteria. The study had the lowest number of GDM cases (5%) as compared to the other included studies (6.5–43%). Finally, for eGWG, all three studies [56,61,62] reported the association between MD adherence and risk of eGWG but differed in study design and analysis. To elaborate, there were the following methodological differences: (1) GWG was a continuous variable and was not categorized into eGWG, making it challenging to determine the association with eGWG [62], whereas another study [61] within the St. Carlos trial, reported inverse associations in relation to MD adherence and eGWG (the GWG variable was categorized into adequate, insufficient, and excessive), and (2) the study used two self-reported, semi-quantitative FFQs as compared to the 7-day food diaries in the de le Torre study [61]; FFQ is primarily directed at food composition largely determined by the combination of foods consumed, but dietary diaries provide more precise quantification of foods consumed and perform better for measuring absolute intakes and predicting energy intakes and subsequent weight gain [74,75].

The MD is suggested to have a protective effect on GDM mediated through a high intake of polyphenols present in key components of the diet, such as extra virgin olive oil, nuts, and fruits and vegetables, by improving insulin sensitivity, lowering glycemic load, activating insulin receptors and stimulating insulin secretion, modulated glucose release resulting in the uptake of glucose in the insulin-sensitive tissues [76], and therefore controlling eGWG. Furthermore, this diet might be attributable to improved inflammation, oxidative stress, and endothelial cell function at a vascular level, thereby lowering high blood pressure typically observed amongst preeclampsia women and increasing placental vascular function and remodeling during early pregnancy [58].

It is well established that maternal nutrition could influence poor birth outcomes, including low birth weight, LGA and SGA deliveries, and finally, the offspring’s metabolic health, through selective epigenetic changes, as supported by preclinical studies [20,77,78,79]. Perhaps antenatal clinics could play a vital role in preventing these poor birth outcomes by providing nutrition education and counseling programs that explain MD’s benefits during pregnancy. In our findings, although most observational studies and RCTs recruited participants during early pregnancy (at around 8–12 weeks) in medical centers, none of them provided nutrition education and counseling focusing on MD benefits during pregnancy at the time of recruitment. Indeed, previous studies have explored the positive effects of nutrition education on healthy and sustainable eating behaviors and lifestyle changes among pregnant women [80,81,82,83,84,85,86]. However, intervention studies are still needed to examine further whether antenatal nutrition education and counseling related to adopting the MD as early as the first trimester can reduce the risk of preeclampsia, GDM, and eGWG. Additionally, we recommend that well-controlled feeding studies evaluate the effect of MD compliance on preeclampsia, GDM, and eGWG during pregnancy by covering the entire gestation period.

### 4.1. Comparison with the Existing Literature

To the best of our knowledge, the current systematic review provides the latest evaluation regarding the evidence on the associations between MD adherence during pregnancy and the risks of preeclampsia, GDM, and eGWG. A systematic review by Marti et al. [38] suggested an inverse association between adherence to MD and risk of GDM and eGWG but reported no findings on preeclampsia. Also, a review by Xu et al. [37] suggested an inverse association between adherence to MD and risk of GDM but reported no association with preeclampsia and did not examine eGWG. Moreover, previous systematic reviews had some methodological differences, including (1) narrow publication range criteria (for instance, 2000–2010) [38], (2) choice of inclusion criteria, including type of population of pregnant women (for example, women who delivered preterm infants or were hypertensive) [87], time of evaluation of the dietary exposures (dietary patterns before or during pregnancy), and choice of study outcomes (maternal or offspring outcomes) [88].

### 4.2. Strengths and Limitations

Our robust methodology appraised the quality and insufficiency of evidence on the association between maternal MD adherence and risk of preeclampsia, GDM, and/or eGWG and identified the requirement to develop well-conducted, highly statistically powered, and longer interventional duration-based studies among pregnant women. All studies included in the present study were carefully examined to meet the eligibility criteria and minimize the potential risk of bias and inclusion error.

The NIH tool was used in this systematic review as it is robust, concise, user-friendly, and efficient in identifying potential flaws in study methodology, measurement of important confounders, inclusion and exclusion criteria, and exposure and outcome measurements. Also, the NIH tool has been widely used in various systematic reviews examining the relationship between diet and diseases [89,90,91,92]. Moreover, the NIH tool did not measure the risk of bias based on a points-based system, but rather on the description of the overall quality of the study, thus making it useful for interpretation.

The Cochrane Risk-of-Bias RoB 2.0 tool was used for RCTs as it is reported to be the most efficient [93], robust, user-friendly, well-established, validated, reliable, and readily accessible for use [93]. Our study used outcome definitions that are widely relevant in clinical settings.

Our systematic review’s findings are widely generalizable and impactful as they represent evidence from several countries, including the USA, Iran, and Mediterranean countries, and were analyzed amongst diverse ethnicities, including Hispanics and African Americans Finally, the study findings reported modest percentages of preeclampsia, GDM, and eGWG cases in the observational studies and RCTs, highlighting the need for more studies.

All four RCTs originated from the same trial and scored a ‘high’ risk of bias on the Cochrane RoB 2.0 tool [60] after poorly scoring against domains such as outcome measurement, randomization process, and outcome reporting.

Furthermore, nutrition interventional trials are limited by several factors, including poor participant compliance, use of different dietary assessment tools (for instance, 24 h dietary recalls, FFQs, food diaries), limited data on the efficacy of nutritional interventions, inadequate information reported by RCTs, small-size study groups with low statistical power, and relatively short duration of follow-up. These limitations might reduce the potential effects or limit the effect size of the associations observed [94].

The RCTs analyzed the effect of single foods, such as extra virgin olive oil and pistachios, over MD adherence during pregnancy [60,61,62]. This could pose two challenges: (1) lower frequency and less compliance for intake of certain single foods in real-world settings, (2) a reductionist approach with a focused effect of a single nutrient or food might be challenging as foods are consumed in combinations [95,96,97].

Finally, nutritional studies are prone to measurement error, underreporting of dietary consumption, and recall bias [98,99]. Our systematic review did not include any non-English languages. However, the excluded studies were thoroughly checked, and no studies were published in non-English languages.

## 5. Conclusions

The overall findings indicated that adherence to MD during pregnancy was inversely associated with GDM and eGWG risks but not with preeclampsia. The systematic review highlighted the scarcity of well-designed and high-sample-sized observational studies and RCTs. Intervention studies are needed to examine whether antenatal nutrition education and counseling related to adopting the MD during pregnancy can improve maternal outcomes and subsequent offspring outcomes, which are strictly intertwined.

## Figures and Tables

**Figure 1 nutrients-17-01723-f001:**
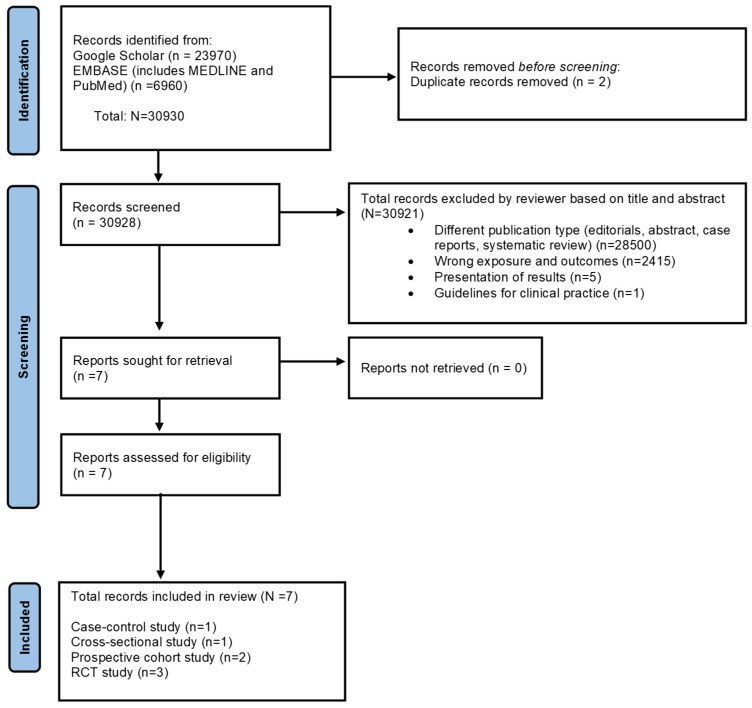
The PRISMA 2020 flow chart shows the study selection process. Identification of studies via databases and registers for the association between Mediterranean diet adherence during pregnancy and the risk of gestational diabetes mellitus, preeclampsia, and excessive gestational weight gain.

**Table 1 nutrients-17-01723-t001:** Eligibility criteria for inclusion of studies.

Parameter	Inclusion Criterion
Participants	Pregnant women
Intervention or exposure	Consumption of Mediterranean diet
Comparison	Any other type of diet
Outcome	Excessive gestational weight gain/preeclampsia/gestational diabetes mellitus
Study design	Randomized controlled trials, observational studies

**Table 2 nutrients-17-01723-t002:** Characteristics of included studies exploring the association between Mediterranean diet adherence during pregnancy and the risk of gestational diabetes mellitus, preeclampsia, and excessive gestational weight gain.

CROSS-SECTIONAL STUDIES								
Serial Number	Author, Year; Country	Period When Study Was Conducted, Recruitment Period of Pregnant Women	Sample Size (N), GDM/Preeclampsia/eGWG Cases (%)	Study Design, Study Population	Dietary Assessment	Exposure: Mediterranean Diet (MD) Score Format Used	Outcome of Interest	Guideline/Method Used to Ascertain eGWG/GDM/ Preeclampsia
1.	Antasouras et al., 2023 [56]; Greece	May 2016 to September 2020, third trimester	N = 5688, Overweight N = 1060 (18.6%), Obese N = 322 (5.7%), GDM N = 372 (6.5%)	Online survey of a general population of pregnant women.	MediDiet questionnaire comprised of 11 food groups. The period when dietary data was recorded is not reported.	MediDiet score ranging 0–55: a posteriori-derived dietary pattern.	Risk of GDM and GWG	**GWG:** WHO method used for GWG **GDM:** Participants’ gestational diabetes diagnoses were recovered from their medical records. A standardized oral glucose tolerance test (OGTT) during gestation was performed, specifically, a fasting OGTT with 75 g of glucose with a cut-off plasma glucose level of >140 mg/dL after 2 h for the first trimester and the following trimester at 24–28 weeks of pregnancy.
**PROSPECTIVE COHORT STUDIES**								
1.	Li et al., 2021 [57]; USA	2009–2013	N = 1887 GDM, N= 85 (5% of N = 1718 women assessed), preeclampsia, N= 61 (3.5% of the N = 1752 women assessed).	Prospective cohort study, general population of pregnant women.	124-item FFQ GDM: Dietary data recorded at 8–13 weeks and 16–22 weeks. Preeclampsia: 8–13 weeks, 16–22 weeks, and 24–29 weeks. Dietary data at 8–13 weeks: Diet History Questionnaire II (modified version) 16–22 weeks, and 24–29 weeks: Automated Self-Administered 24-h (ASA24) Dietary Assessment Tool.	Adherence to Mediterranean Diet by the aMED score ranging from 0 to 9.	Risk and severity of GDM and preeclampsia.	**GDM:** Gestational diabetes was defined by women’s oral glucose challenge test results using the Carpenter-Coustan criteria (at least 2 values met or exceeded: fasting—95 mg/dL, 1 h—180 mg/dL, 2 h—155 mg/dL, 3 h—140 mg/dL), and/or by receipt of GDM medications. **Preeclampsia:** The 2002 ACOG criteria defined preeclampsia is a new onset of elevated blood pressure (≥140 mm Hg or a diastolic blood pressure ≥ 90 mm Hg) after 20 weeks with proteinuria (≥0.3 g of protein in a 24 h urine specimen).
2.	Minhas et al., 2022 [58]; USA	1998–2016 Maternal age: 28 (23–33 y) Mixed race/ethnicity, majority black: (4030/47%), Hispanic: (2423/28%).	N = 8507, Preeclampsia N = 848 (10%)	Prospective cohort study, participants recruited from a medical center.	16-item FFQ Dietary data was recorded after 24–72 h of delivery and covered the dietary intake during pregnancy.	Mediterranean-style diet score (MSDS) (4–38)	Risk of preeclampsia.	Preeclampsia included in any form, including mild or severe preeclampsia, eclampsia, or HELLP (hemolysis, elevated liver enzymes, low platelet count) syndrome.
**CASE-CONTROL STUDIES**								
1.	Izadi et al., 2016 [59]; Iran	No year indicated, between 5 and 28 weeks of gestation.	N = 463, N of GDM cases = 200 (43%)	Hospital-based case-control study, general population of pregnant women.	Three 24 h dietary recalls. Dietary data recorded between 5 and 28 weeks of pregnancy.	MedDiet score ranging 0–9 by Trichopoulou et al. [31]	Risk of GDM	GDM was ascertained if the pregnant women had abnormal fasting glucose (FG; >95 mg/dL or 1-h postprandial glucose > 140 mg/dL for the first time in pregnancy).
**RANDOMIZED CONTROLLED TRIALS**								
**Serial Number**	**Author, Year; Country**	**Period When Study Was Conducted, Recruitment Period of Pregnant Women**	**Sample Size (N), Control Group (CG) (n)/Intervention Group (IG) (n), GDM/Preeclampsia/eGWG cases (N) (%), (n/CG, n/IG)**	**Control Group Diet**	**Dietary Assessment**	**Exposure:** **Mediterranean Diet (MD) Score Format Used for Interventional Group**	**Outcome of Interest**	**Guideline/Method Used to Ascertain eGWG/GDM/** **Preeclampsia**
1.	Assaf-Balut et al., 2017 [60]; Spain	January–December 2015, 8–12 weeks of pregnancy.	N = 874, 440/434 GDM N = 177 (20.2%), (103/440, 74/434) GDM was distributed at random between two control and interventional groups.	A standard diet with limited fat intake.	7-day food diaries Dietary data recorded during 8–12 weeks of pregnancy.	14-point Mediterranean Diet Adherence Screener (MEDAS). MedDiet supplemented with a recommendation of a daily consumption of at least 40 mL of EVOO and a handful (25–30 g) of pistachios.	Risk of GDM	IADPSG criteria were used to diagnose GDM at 24 ± 28 GW with a single 2 h 75 g oral glucose tolerance test.
2.	de la Torre et al., 2019 [61]; Spain	January–November 2017, 8–12 weeks of pregnancy	N = 932 GDM N = 130 (13.9%), (Non-GDM control group = 802/GDM interventional group = 130) **GWG:** Excessive, N = 405/932 (43.4%) NGT group: 371/802, (46.3%) GDM group: 34/130 (26.2%) Adequate GWG, N = 357/932 (38.3%) NGT group: 296/802, (36.9%) GDM group: 61/130 (46.9%) Insufficient GWG, N = 170/932 (18.2%) NGT group: 135/802, (16.8%) GDM group: 35/130 (26.9%) **Preeclampsia:** N = 10/932 (1%) NGT group: 9/802, (1.1%) GDM group: 1/130 (0.8%)	No control diet administered. All participants were educated regarding Mediterranean diet guidelines with exclusive consumption of EVOO, and a daily handful of nuts (not provided).	7-day food diaries Dietary data recorded during 8–12 weeks of pregnancy.	14-point Mediterranean Diet Adherence Screener (MEDAS). Education on the implementation of Mediterranean diet guidelines supplemented with exclusive consumption of EVOO, and a daily handful of nuts (not provided).	Risk of excessive GWG and preeclampsia.	**GDM:** IADPSG and WHO 2013 criteria was used to diagnose GDM at 24 ± 28 GW with a single 2-h 75-g oral glucose tolerance test. **eGWG:** eGWG was defined as a gestational weight gain 3 Kg above the designated target according to pre-gestational BMI. **Preeclampsia:** >140 mmHg systolic/90 mmHg diastolic with proteinuria > 300 mg in 24 hr after 20 gestational weeks.
3.	Melero et al., 2020 [62]; Spain	2016–2017, 8–12 weeks of pregnancy	N = 600 CG: 142 (23.6%), IG: 143 (23.8%) Real world group (RWG): 315 (52.5%) GDM N = 91/544 (16.7%) CG N = 34/132 (25.8%), IG N = 19/128 (14.8%), RWG N = 38/284 (13.4%) Preeclampsia N = 15 (2.7%) CG N = 6/132 (4.5%), IG N = 5/128 (3.9%), RWG N = 4/284 (1.4%) GWG outcome was not categorized in the study.	CG was advised to restrict fat intake, with the consumption of extra virgin olive oil (EVOO) limited to a maximum of 40 mL/day, and nuts < 3 days per week as usually recommended.	Two semi-quantitative FFQ Dietary data recorded during 8–12 weeks of pregnancy.	14-point Mediterranean Diet Adherence Screener (MEDAS). MedDiet supplemented with the recommendation of daily consumption of at least 40 mL of EVOO and a handful (25–30 g) of pistachios at least 3 days a week.	Risk of GDM, excessive GWG, and preeclampsia.	**GDM:** IADPSG criteria were used to diagnose GDM at 24 ± 28 GW with a single 2-h 75 g oral glucose tolerance test. **GWG:** No evidence of an association. **Preeclampsia:** >140 mmHg systolic/90 mmHg diastolic with proteinuria > 300 mg in 24 hr after 20 gestational weeks.

**Table 3 nutrients-17-01723-t003:** Results of the included studies exploring the association between Mediterranean diet adherence during pregnancy, and the risk of gestational diabetes mellitus, preeclampsia, and excessive gestational weight gain.

CROSS-SECTIONAL STUDIES						
Serial Number	Author, Year	Outcome of Interest	Covariates	Statistical Methods	Results	Conclusion
1.	Antasouras et al., 2023 [56]; Greece	Risk of GDM and eGWG	Maternal age, educational and economic status, nationality, type of residence, smoking habits, parity, pre-pregnancy BMI status, preterm birth, gestational diabetes, gestational hypertension, type of delivery, and exclusive breastfeeding.	A multivariate binary logistic regression analysis was applied to evaluate whether compliance with the MD may exert an independent impact on sociodemographic and anthropometric parameters, perinatal outcomes, and breastfeeding practices.	Decreased adherence to MD was associated with an increased risk of eGWG (OR: 1.78; 95% CI 1.51 to 2.02) and GDM (OR: 2.32; 95% CI 2.13 to 2.57).	The MedDiet score was inversely associated with GDM risk and excessive eGWG.
**PROSPECTIVE STUDIES**						
1.	Li et al., 2021 [57]; USA	Risk of GDM, preeclampsia, and common pregnancy complications	Maternal age, race (non-Hispanic white, non-Hispanic black, Hispanic, Asian), education (<high school, high school, some college, bachelor, graduate), marriage/cohabiting (yes, no), nulliparity (yes, no), pre-pregnancy BMI (kg/m^2^), family history of diabetes (yes, no), light to vigorous physical activities (hour/week, sleep durations (5–6, 7, 8–9, 10+ h/day), and total energy intake (kcal/day).	Log-binomial regression models to explore associations between aMED adherence scores and risk of GDM.	**GDM:** High aMED score adherence was not associated with a lower risk of GDM during 8–13 weeks of pregnancy (Q4 vs. Q1: RR 0.61; 95% CI: 0.25 to 1.48) and during 16–22 weeks of pregnancy(Q4 vs. Q1: RR 0.61; 95% CI: 0.33 to 1.15). **Preeclampsia:** High aMED score adherence was not associated with a lower risk of preeclampsia during 8–13 weeks of pregnancy (Q4 vs. Q1: RR 0.68; 95% CI: 0.25 to 1.85), during 16–22 weeks of pregnancy. (Q4 vs. Q1: RR 0.67; 95% CI: 0.34 to 1.32) and during 24–29 weeks of pregnancy (Q4 vs. Q1: RR 0.47; 95% CI: 0.18 to 1.21).	**GDM:** High MD adherence was not associated with a lower risk of GDM. **Preeclampsia:** High MD adherence was not associated with a lower risk of preeclampsia.
2.	Minhas et al., 2022 [58]	Risk of preeclampsia	Age (categorical: <21, 21–30, ≥30 years), race and ethnicity (White, Black, Hispanic, and other), education (categorical: 1 = no, school/ elementary school, 2 = high school, 3 = some college or above), marital status (categorical: 1 = married, 2 = unmarried, 3 = unknown), smoking status (binary: 0 = never smoker during pregnancy, 1 = smoking during pregnancy), parity (binary: 0 = nulliparous, 1 = parous), and pre-pregnancy obesity (binary: 0 = body mass index < 30 kg/m^2^, 1 = body mass index ≥ 30 kg/m^2^).	Multivariable adjusted logistic regression models were used to evaluate the association between adherence to the Mediterranean-style diet, and the risk of preeclampsia.	Compared with women who scored in the lowest tertile of the MSDS, women in the middle tertile (OR 0.72; 95% CI: 0.59 to 0.89) and highest tertile had (OR 0.78; 95% CI: 0.64 to 0.96) were inversely associated with lower odds of preeclampsia.	Possible protective effect of the Mediterranean-style diet on risk of preeclampsia.
**CASE-CONTROL STUDIES**						
1.	Izadi et al., 2016 [59]	Risk of GDM	Age, energy, number of children, and socioeconomic status.	Multiple logistic regression models to assess MD adherence and the risk of GDM.	Inverse association between high MedDiet score and the risk of GDM (Tertile 3 vs. tertile 1: OR: 0.20, 95% CI 0.50 to 0.70).	An inverse association between the MedDiet score and the risk of GDM.
**RANDOMIZED CONTROLLED TRIALS**						
1.	Assaf-Balut et al., 2017 [60]; Spain	Risk of GDM	Age, ethnicity, parity, BMI (continuous), gestational, personal and family history, and smoker (categorical: never, current, former smoker).	Logistic regression analyses were used to assess the effect of the intervention on the risk of GDM.	Inverse association between high MEDAS score supplemented with extra virgin olive oil and pistachios, and the risk of GDM (OR: 0.75, 95% CI 0.57 to 0.98) in the intervention group.	Potential protective effect of high Mediterranean diet adherence, extra virgin olive oil, and pistachios on risk of GDM.
2.	de la Torre et al., 2019 [61]; Spain	Risk of eGWG and preeclampsia, and adverse maternal-fetal outcomes.	None.	Logistic binary regression analyses were used to assess the effect of GDM on eGWG and preeclampsia.	Amongst GDM women, a high adherence to Mediterranean diet score was associated with a lesser risk of eGWG, (RR: 0.91, 95% CI 0.86 to 0.96). However, it was not associated with a lesser risk of preeclampsia.	Potential protective effect of high Mediterranean diet adherence amongst GDM women on eGWG, but not preeclampsia.
3.	Melero et al., 2020 [62]; Spain	Risk of GDM, GWG and preeclampsia, and other adverse maternal-fetal events.	Age, parity, and BMI.	Logistic regression was used to assess the effect of the MD nutritional therapy for the GDM, GWG, and preeclampsia.	Participants with higher adherence to MD plus EVOO and pistachios intervention were associated with lower risk of GDM (RR: 0.72, 95% CI 0.50 to 0.97) in the IG and (RR: 0.77, 95% CI 0.61 to 0.97) in the RWG, respectively. However, it is not associated with preeclampsia.	High adherence to the Mediterranean diet was associated with a lower risk of GDM but not preeclampsia and GWG.

## Data Availability

The data described in the manuscript, code book, and analytic code will not be made available because it is only a systematic review of published studies.

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
