# Peer review of "Impact of Mediterranean Diet Adherence During Pregnancy on Preeclampsia, Gestational Diabetes Mellitus, and Excessive Gestational Weight Gain: A Systematic Review of Observational Studies and Randomized Controlled Trials"

_nutrients, 2025, doi:10.3390/nu17101723_

Round 1

Reviewer 1 Report

Comments and Suggestions for Authors

I believe the title is excessively long and redundant. A systematic review is classified as secondary research because it relies on existing studies rather than collecting new data. This distinction should clear up any confusion. Primary sources provide direct access to the research subject, while secondary sources offer second-hand information and commentary from other researchers. Examples of secondary sources include journal articles, reviews, and academic books. For this reason, the PRISMA guidelines specify that only primary studies should be included in a systematic review. Thus, having such an overly lengthy title seems unnecessary.  On the positive side, the review effectively addresses whether the maternal Mediterranean diet has an impact on preeclampsia, gestational diabetes mellitus, and excessive gestational weight gain.

The criteria now referred to as “criteria for the PICOS” should be recognized as eligibility criteria and should follow the PICO framework. Consecutively, Table 1 provides an overview of the study characteristics; however, it should not be considered a "PICO table." Among the exclusion criteria, animal models were listed, but human studies were notably absent from the inclusion criteria. Furthermore, it is unclear whether this review included laboratory (in vitro) studies and preprint publications.

Distinguishing between results and methodology in a systematic review can sometimes be challenging, particularly in section 2.6. Synthesis of Results. It's easy to confuse this as the systematic review provides a summary of data from the results of multiple individual studies. 

Author Response

REBUTTAL SHEET

Dear Reviewers

Thank you for the time taken to provide comments on our manuscript. We have now addressed each of the comments (in red).

Reviewer 1

I believe the title is excessively long and redundant. A systematic review is classified as secondary research because it relies on existing studies rather than collecting new data. This distinction should clear up any confusion. Primary sources provide direct access to the research subject, while secondary sources offer second-hand information and commentary from other researchers. Examples of secondary sources include journal articles, reviews, and academic books. For this reason, the PRISMA guidelines specify that only primary studies should be included in a systematic review. Thus, having such an overly lengthy title seems unnecessary.  On the positive side, the review effectively addresses whether the maternal Mediterranean diet has an impact on preeclampsia, gestational diabetes mellitus, and excessive gestational weight gain.

  • Thank you for your comment, yes, indeed it is slightly long. However, during the registration process at PROSPERO, we chose to keep the title specific to the maternal outcomes to be clear and easily identifiable through citation searches in databases such as Scopus and EMBASE.

The criteria now referred to as “criteria for the PICOS” should be recognized as eligibility criteria and should follow the PICO framework. Consecutively, Table 1 provides an overview of the study characteristics; however, it should not be considered a "PICO table."  Among the exclusion criteria, animal models were listed, but human studies were notably absent from the inclusion criteria. Furthermore, it is unclear whether this review included laboratory (in vitro) studies and preprint publications.

  • Thank you very much for these comments, we have now included the changes in red under section 2.1, page 2 and 3.

Distinguishing between results and methodology in a systematic review can sometimes be challenging, particularly in section 2.6. Synthesis of Results. It's easy to confuse this as the systematic review provides a summary of data from the results of multiple individual studies.

  • Thank you for this comment, we agree and so we omitted section 2.6, and added the following line in the Results section 3.3: “Finally, the findings were synthesized according to each study design, and results from only models fully adjusted for covariates were used (refer to Table 3 for the list of covariates used in each study).”

Reviewer 2 Report

Comments and Suggestions for Authors

This study uses meta-analysis to explore the impact of adherence to the Mediterranean diet during pregnancy on preeclampsia, gestational diabetes mellitus, and excessive gestational weight gain. A similar meta-analysis was published in the past three years, which diminishes the novelty of this paper. Additionally, issues regarding writing standardization need to be addressed.

As a meta-analysis, why were forest plot analysis not conducted? For example, subgroup analyses should be performed for preeclampsia, gestational diabetes mellitus, and excessive gestational weight gain.

The introduction does not explain why the effects of the Mediterranean diet on indicators such as preeclampsia, gestational diabetes mellitus, and excessive gestational weight gain were studied. Is it simply because previous studies did not report on them?

The data analysis in this study is insufficient. The authors merely describe the results of individual studies without accounting for differences in sample sizes, which affects the weight of each study. Further analysis is needed to better present the results.

Section 3.3: Shouldn’t the interpretation of the results be placed in the discussion section?

The Minhas study involved a very large survey population. If this large-scale study yielded results different from others, what does this imply?

The number of references is excessive. A simple meta-analysis does not require this many references. It is unclear why the authors included so many; they should be filtered to retain only the most relevant ones. I also did not see the authors engage in extensive discussion, so including such a large number of references is unreasonable.

The discussion section of this meta-analysis is weak. First, maternal blood sugar issues and diabetes can lead to severe consequences, so the benefits of the Mediterranean diet should be emphasized. Additionally, the explanation for the positive effects of the Mediterranean diet is insufficient.

Beyond polyphenol intake, dietary fiber, for example, also benefits gestational diabetes and excessive weight gain. Furthermore, merely attributing the effects to cardiovascular inflammation and oxidative stress is inadequate.

Section 4.1: By comparing with past literature, the main difference in this study is the inclusion of a few additional indicators, which does not constitute significant novelty.

Figure captions need only be written once below the respective figures.

The reference format is incorrect and does not comply with MDPI’s guidelines.

Author Response

REBUTTAL SHEET

Dear Reviewers

Thank you for the time taken to provide comments on our manuscript. We have now addressed each of the comments (in red).

Reviewer 2

This study uses meta-analysis to explore the impact of adherence to the Mediterranean diet during pregnancy on preeclampsia, gestational diabetes mellitus, and excessive gestational weight gain. A similar meta-analysis was published in the past three years, which diminishes the novelty of this paper. Additionally, issues regarding writing standardization need to be addressed.

As a meta-analysis, why were forest plot analysis not conducted? For example, subgroup analyses should be performed for preeclampsia, gestational diabetes mellitus, and excessive gestational weight gain.

The data analysis in this study is insufficient. The authors merely describe the results of individual studies without accounting for differences in sample sizes, which affects the weight of each study. Further analysis is needed to better present the results.

  • Thank you for stating these comments. We appreciate the comments and would like to respectfully clarify that our overarching aim was to design and conduct a systematic review only, with the sole objective of providing the most up-to-date evidence regarding the three maternal outcomes during pregnancy. Therefore, since the study conceptualization, we registered this study only as a systematic review at PROSPERO after recognizing methodological challenges related to meta-analyzing the MD (as it is often recorded in multiple ways around the world), and different standards used to assess the maternal outcomes. Thus, citing these reasons we decided to conduct only a systematic review and not analyze the results using various techniques required for meta-analysis. We also acknowledge that there has been a similar meta-analysis published recently in 2024, and we discussed the paper in our systematic review. Finally, regarding the issue of novelty, our systematic review provided an overview of the existing literature and has made recommendations for future research, ultimately which could contribute towards clinical application and decision-making.

The introduction does not explain why the effects of the Mediterranean diet on indicators such as preeclampsia, gestational diabetes mellitus, and excessive gestational weight gain were studied. Is it simply because previous studies did not report on them?

  • Thank you for this comment. To address your comment, we have made the following changes to the text (highlighted in yellow): “However, previous reviews reported limited (either of the maternal outcomes) and inconclusive findings on the association between MD adherence and the risks of preeclampsia, GDM, and eGWG due to methodological inconsistencies and study limitations [37,38]. This highlighted the requirement of clear and comprehensive systematic reviews that provide up-to-date evidence for a general population of well-nourished pregnant women useful in clinical and public health settings [51–54]. We conducted a systematic review of observational studies and randomized controlled trials (RCTs) to examine the association between MD adherence during pregnancy and preeclampsia, GDM, and/or eGWG.”

Section 3.3: Shouldn’t the interpretation of the results be placed in the discussion section?

  • Thank you for the comment, we recognize this issue and have incorporated the changes in the Discussion section accordingly.

The Minhas study involved a very large survey population. If this large-scale study yielded results different from others, what does this imply?

  • Thank you for the comment. We have now included a sentence (highlighted in yellow) in the text: “Among the four studies that examined preeclampsia as a study outcome, only one study [69] reported an inverse association between high MD adherence during pregnancy and risk of preeclampsia. This might be due to three reasons: (1) different MD adherence scales used— RCTs used MEDAS tool as compared to the MSDS tool used in the observational study; (2) preeclampsia cases were more frequent in the observational study (10%) as compared to the RCTs (1-3%), increasing the power to detect associations in the regression models [73,74]; and (3) the duration of the intervention in the RCT could have been too short to observe any effects. Therefore, large sample-sized studies using a consistent definition of outcome could yield more evidence in relation to preeclampsia.”

The number of references is excessive. A simple meta-analysis does not require this many references. It is unclear why the authors included so many; they should be filtered to retain only the most relevant ones. I also did not see the authors engage in extensive discussion, so including such a large number of references is unreasonable.

  • Thank you for your comments, respectfully, we highlighted that we conducted only a systematic review and not a meta-analysis of a very well-researched topic. Therefore, we deemed it necessary to identify previous literature and cite their contribution to the large scheme of evidence for this research topic to be driven forward (pertaining to the maternal outcome indicators). Hence, we agree the references appear extensive, but the references are also for other sections of the paper, such as methodology, introduction, and discussion where we refer to others’ works to present an overall review.

The discussion section of this meta-analysis is weak. First, maternal blood sugar issues and diabetes can lead to severe consequences, so the benefits of the Mediterranean diet should be emphasized. Additionally, the explanation for the positive effects of the Mediterranean diet is insufficient.

Beyond polyphenol intake, dietary fiber, for example, also benefits gestational diabetes and excessive weight gain. Furthermore, merely attributing the effects to cardiovascular inflammation and oxidative stress is inadequate.

  • Thank you for your comments. To address these important concerns, we would like to state that we have already discussed the positive impact of MD on the mediation through maternal indicators such as GDM, eGWG, and finally preeclampsia, and also hypothesized their role by briefly explaining the mechanistic pathways aspects in detail.

“The MD is suggested to have a protective effect on GDM mediated through a high in-take of polyphenols present in key components of the diet such as extra virgin olive oil, nuts, and fruits and vegetables by improving insulin sensitivity, lowering glycemic load, activating insulin receptors and stimulating insulin secretion, modulated glucose release resulting in the uptake of glucose in the insulin-sensitive tissues (77), and therefore control-ling eGWG. Further, this diet might be attributable to improved inflammation, oxidative stress, and endothelial cell function at a vascular level, thereby lowering high blood pres-sure typically observed amongst preeclampsia women and increasing placental vascular function and remodeling during early pregnancy (69).”

Section 4.1: By comparing with past literature, the main difference in this study is the inclusion of a few additional indicators, which does not constitute significant novelty.

  • Thank you for your comment, we do agree that this is a well-known, well-researched topic that could diminish the novelty of our study. However, our systematic review has provided the most recent update of the existing literature related to the selected maternal indicators and discussed some important recommendations for future studies—and could have clinical significance.

Figure captions need only be written once below the respective figures.

  • Thank you, it is not there in the current manuscript version.

The reference format is incorrect and does not comply with MDPI’s guidelines.

  • Thank you for this and we sincerely apologize. We have now made changes in the manuscript.

………………………….

Round 2

Reviewer 1 Report

Comments and Suggestions for Authors

In my opinion, the revisions are sufficient.

Reviewer 2 Report

Comments and Suggestions for Authors

The author has made changes to my questions.